# HOPE: Shape Matching Via Aligning Different K-hop Neighbourhoods

**Barakeel Fanseu Kamhoua[1], Huamin Qu[1]\*,**
[1]The Hong Kong University of Science and Technology

## Abstract

Accurate and smooth shape matching is very hard to achieve. This is because for accuracy, one needs unique descriptors (signatures) on shapes that distinguish different vertices on a mesh accurately while at the same time being invariant to deformations. However, most existing unique shape descriptors are generally not smooth on the shape and are not noise-robust thus leading to non-smooth matches. On the other hand, for smoothness, one needs descriptors that are smooth and continuous on the shape. However, existing smooth descriptors are generally not unique and as such lose accuracy as they match neighborhoods (for smoothness) rather than exact vertices (for accuracy). In this work, we propose to use different k-hop neighborhoods of vertices as pairwise descriptors for shape matching. We use these descriptors in conjunction with local map distortion (LMD) to refine an initialized map for shape matching. We validate the effectiveness of our pipeline on benchmark datasets such as SCAPE, TOSCA, TOPKIDS, and others.

## 1   Introduction

Shape matching is a very important task and has been increasingly so with the increase in the availability and affordability of 3D scans [31]. It has important applications including but not limited to shape registration [2, 37], comparison [1, 13], recognition [6], and retrieval [50].

Shapes can undergo different types of transformations which we will group as isometric and non-isometric transformations in this work. Nonrigid isometric transformations preserve most of the geometric properties of the shape i.e., after the transformation, its angles, geodesic distances, scale, connectivity, and other geometric properties are mostly preserved. On the other hand, most geometric properties are not preserved under nonrigid non-isometric transformations. Examples of isometric transformations are rotations and translations (though minute scaling may also be included in this category for nonrigid cases). While examples of non-isometric transformations are splitting, scaling, dilating, and others which change the geometric properties of the shapes.

Several works have been proposed to address shape matching. However, most of these methods will either aim for smoothness [12, 17, 19, 13, 1, 34, 39, 37, 44, 31, 20] or for accuracy [45], but not both. Moreover, most of these methods will focus on specific types of deformations undergone by the shape either isometric [12, 19, 13, 1, 34, 39, 37, 44, 31, 20, 49] or non-isometric [14, 22, 52]. Some methods such as 2D-GEM [22] have also tried to address both accuracy and smoothness in shapes undergoing both isometric and non-isometric deformations. However, 2D-GEM is very parameter-dependent, using different user-defined parameters for isometric shapes than for non-isometric shapes which in real life is not practical as one usually does not know whether one is dealing with isometric or non-isometric deformations.

Given these obsevations, there is a need for a pipeline that aims for accuracy and smoothness across different settings without significantly changing the pipeline. In this light, we propose HOPE, a k-hop

---

\*Corresponding author: huamin@cse.ust.hk

38th Conference on Neural Information Processing Systems (NeurIPS 2024).

neighborhood-based refinement technique. The rest of the paper is organized into; (a) preliminaries, (b) related work, (c) HOPE, and (d) experiments, (e) limitations and remarks, and (f) conclusion.

## 2 Preliminaries

3D shapes are usually represented by their coordinates $\mathcal{X} \in \mathbb{R}^{n \times 3}$ and their triangulations (meshes) which can be used to build an adjacency $\mathcal{A} \in \mathbb{R}^{n \times n}$, with $\mathcal{A}(i, j) = 1$ if vertices $i$ and $j$ are connected in the shape and $0$ otherwise.

Given two three-dimensional shapes $\mathcal{M}$ and $\mathcal{N}$ with $n_{\mathcal{M}}$ and $n_{\mathcal{N}}$ vertices respectively. Though our proposed pipeline (HOPE) and some other baselines work with $n_{\mathcal{M}} \neq n_{\mathcal{N}}$, we will assume $n_{\mathcal{M}} = n_{\mathcal{N}} = n$, and use $n$ for simplicity except when specified. The goal of nonrigid shape matching is to find a meaningful correspondence $\mathcal{T} : \mathcal{M} \to \mathcal{N}$, where $\mathcal{T}$ is (a) bijective, (b) continuous (smooth) , and (c) similar vertices should be matched to each other [26, 35, 22]. A measure that captures both the smoothness and the accuracy of the map is the geodesic error. The geodesic error measures how far a map $\mathcal{T}$, maps a vertex from its matching position given by the ground truth map $\hat{\mathcal{T}}$, where a higher map accuracy will correspond to a higher proportion of vertices having geodesic error $0$, and a smoother map will correspond to a huge increase in the geodesic error curve as we move slightly away from error $0$ (implying that several vertices though not accurate are mapped to their intended neighborhoods).

The two main components of shape matching involve: (1) initializing a map, and (2) refining the initialized map.

First, the map is initialized either by using landmarks vertices (i.e., vertices with known correspondences) or by using robust descriptors. These descriptors can be based on: (a) the spectrum of the shape laplacian [8, 4, 33], (b) The face normals, vertex location, and triangulation [45, 41], geodesic distances [44, 1] or others [14, 46, 15, 16, 11, 43, 23]. In this work, we will assume that one such initialization approach has been utilized and we have an initial map at hand.

Second, the initialized map $\mathcal{T}^0$ can then be refined by either using pair-wise descriptors [36, 26, 52, 48, 14, 38, 22] and solving:

$$\mathcal{T}^t = \arg\min_{\mathcal{T}^t} ||\mathcal{W}_{\mathcal{M}}(\mathcal{T}^t, \mathcal{T}^{t-1}) - \mathcal{W}_{\mathcal{N}}||, \tag{1}$$

or by using vertex-wise descriptors [2, 34, 39, 37, 31, 26, 35, 19, 20, 44, 1, 28] and solving:

$$\mathcal{T}^t = \arg\min_{\mathcal{T}^t} ||\mathcal{Q}_{\mathcal{M}}(\mathcal{T}^t, :) - \mathcal{Q}_{\mathcal{N}} f(\mathcal{T}^{t-1})||, \tag{2}$$

Where in equation 1, $\mathcal{W}_{\mathcal{M}}(\mathcal{T}^t, \mathcal{T}^{t-1}) \in (R)^{n \times n}$ is the pair-wise descriptors of vertices for shape $\mathcal{M}$ with its rows and columns aligned using the map $\mathcal{T}^t$ and the previous iterations map $\mathcal{T}^{t-1}$ respectively. $\mathcal{W}_{\mathcal{N}}$ is the pair-wise descriptor for shape $\mathcal{N}$. Here for conciseness for aligning rows or columns we will use $\mathcal{T}$ (while in reality if the row map is $\mathcal{T} = argmax(\mathcal{P}, \text{dim=-1})$ the column should be $\mathcal{T}' = argmax(\mathcal{P}, \text{dim=-2})$ where $P$ is the permutation matrix such that $\mathcal{P}\mathcal{Q}_{\mathcal{M}} = \mathcal{Q}_{\mathcal{N}}$ and $\mathcal{P}\mathcal{W}_{\mathcal{M}}\mathcal{P}^T = \mathcal{W}_{\mathcal{N}}$

While in equation 2, $\mathcal{Q}_{\mathcal{M}}(\mathcal{T}^t, :) \in \mathbb{R}^{n \times d}$ is the vertex-wise descriptors for shape $\mathcal{M}$ with feature dimension $d$ for each vertex and with its rows aligned according to the map $\mathcal{T}^t$, $f(\mathcal{T}^{t-1})$ is a function (such as functional map[34]) to transfer the previous iteration's map to the descriptor space, and $\mathcal{Q}_{\mathcal{N}} \in \mathbb{R}^{n \times d}$ is the vertex-wise descriptors for shape $\mathcal{N}$.

For the map $\mathcal{T}$ refined by equations 1 or 2 to have a high accuracy (in the ideal case), the pair-wise descriptors $\mathcal{W}_{\mathcal{N}}(i, :) \in \mathbb{R}^{1 \times n}$ or vertex-wise descriptors $\mathcal{Q}_{\mathcal{N}}(i, :) \in \mathbb{R}^{1 \times d}$ will need to be: (a) unique for each vertex $i$ on shape $\mathcal{N}$, and (b) identical to its ground truth corresponding pair-wise feature $\mathcal{W}_{\mathcal{M}}(\hat{\mathcal{T}}(i), :) \in \mathbb{R}^{1 \times n}$ or vertex-wise feature $\mathcal{Q}_{\mathcal{M}}(\hat{\mathcal{T}}(i), :)$ on shape $\mathcal{M}$. Where $\hat{\mathcal{T}}$ is the ground truth map. Moreover, for smoothness, vertices should be mapped such that they remain in their relative neighborhoods before and after the mapping.

## 3 Related Work

ZoomOut [31] and other related works[19, 13, 1, 34, 39, 37, 26, 35] all aim for smoothness of the map and focus on the settings in which the shapes have undergone an isometric deformation, such

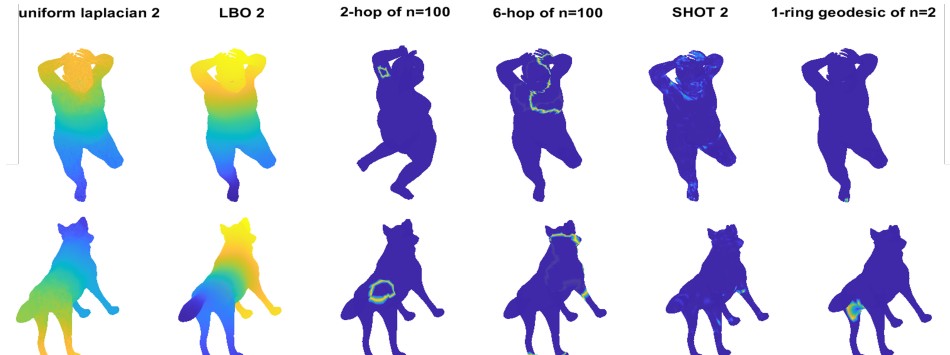

| uniform laplacian 2 | LBO 2 | 2-hop of n=100 | 6-hop of n=100 | SHOT 2 | 1-ring geodesic of n=2 |

Figure 1: For sample shapes from the TOPKIDS (first row) and TOSCA (second row), this figure shows the second eigenvector of the mesh laplacian (LBO 2), the second eigenvector of the uniform shape laplacian from the triangulation adjacency (uniform shape laplacian 2), the 2-hop and 6-hop neighborhoods of vertex 100, the second SHOT descriptor and the LMD of vertex 2.

that geometric shape properties are preserved. These works use vertex-wise descriptors usually based on some truncated (reduced) basis of specific shape properties, generally leading to some loss of information such as uniqueness. For example, consider the $d$ dimensional vector $\mathcal{U}(i,:) \in \mathbb{R}^d$ as the vertex-wise descriptors of vertex $i$, supposing that the matrix $\mathcal{U} \in \mathbb{R}^{n \times d}$ contains the first $d$ eigenvectors of the uniform shape laplacian built from $\mathcal{A}$, it can be seen that $\mathcal{U}(i,:)$ is the soft cluster assignment of vertex $i$, and as such most vertices in the same cluster as $i$ will have similar vertex-wise descriptors. This can be seen in figure 1 where the second eigenvector $\mathcal{U}(:,2) \in \mathbb{R}^n$ is shown for example shapes from TOSCA and TOPKIDS. One can see that $\mathcal{U}(:,2) \in \mathbb{R}^n$ is indeed a soft cluster assignment grouping some vertices in the same cluster together. This explains the success in achieving smoothness by some methods that use the $d$ dimensional spectrum of the shape laplacian.

**Theorem 3.1** *Given the shape descriptor $\mathcal{U}(i,:) \in \mathbb{R}^d$ as the vertex-wise descriptors of vertex $i$, supposing that the matrix $\mathcal{U} \in \mathbb{R}^{n \times d}$ contains the first $d$ eigenvectors or left singular vectors of some unique shape pairwise descriptor $\mathcal{W}$. Using $\mathcal{W}$ for the map refinement via Functional maps helps group nearby clusters together assuming the functional map is perfectly accurate.*

**Proof 3.1** *Recall that given a map $\mathcal{T}^{t-1}$, and two vertex-wise descriptors $\mathcal{U}_{\mathcal{M}}$ and $\mathcal{U}_{\mathcal{N}}$ both $\in R^{n \times d}$ the functional map $\mathcal{C} \in R^{d \times d}$ is obtained as:*

$$\mathcal{C}^t = \min_{\mathcal{C}^t} ||\mathcal{U}_{\mathcal{M}}(\mathcal{T}^{t-1},:) - \mathcal{U}_{\mathcal{N}}\mathcal{C}^t||$$

*This functional map can then be used to convert functions in the basis of shape $\mathcal{M}$ into those of shape $\mathcal{N}$ and vice versa. This is then used to refine the map via solving equation 2 where $f(\mathcal{T}^t) = \mathcal{C}^t$. One can see that $\mathcal{U}_{\mathcal{N}}\mathcal{C}^t$ converts the soft clusters (left singular vectors or first eigenvectors) of shape $\mathcal{U}_{\mathcal{N}}$ into the basis space of the descriptors (soft clusters) of shape $\mathcal{U}_{\mathcal{M}}$, and so solving 2 is basically aligning the soft clusters in the same basis space. Thus solving equation 2 is matching aligned soft clusters.*

In fact, methods such as ZoomOut [31, 20, 34] which iteratively refine the map $\mathcal{T}$ by using an increasing number of eigenvectors of the shape laplacian can be said to be refining the map by increasingly aligning more fine-grained clusters of the shapes. Figure 2 shows that though ZoomOut's recovered maps $\mathcal{T}$ on the isometric SCAPE[26] dataset are generally smooth (seen by the rapid increase of the geodesic curve), they are generally not accurate due to the non-uniqueness of the descriptors it uses to refine the initialized map (seen by the proportion of vertices at geodesic error 0).

Moreover, though geometric properties may be preserved even for some mild non-isometric deformations, they usually are not preserved when the transformation deviates significantly from isometry. For example, consider the shape Laplace Beltrami Operator (LBO) on which many vertex-wise and pair-wise descriptors are built:

$$\mathcal{L} = \mathcal{V}^{-1}\mathcal{S} \tag{3}$$

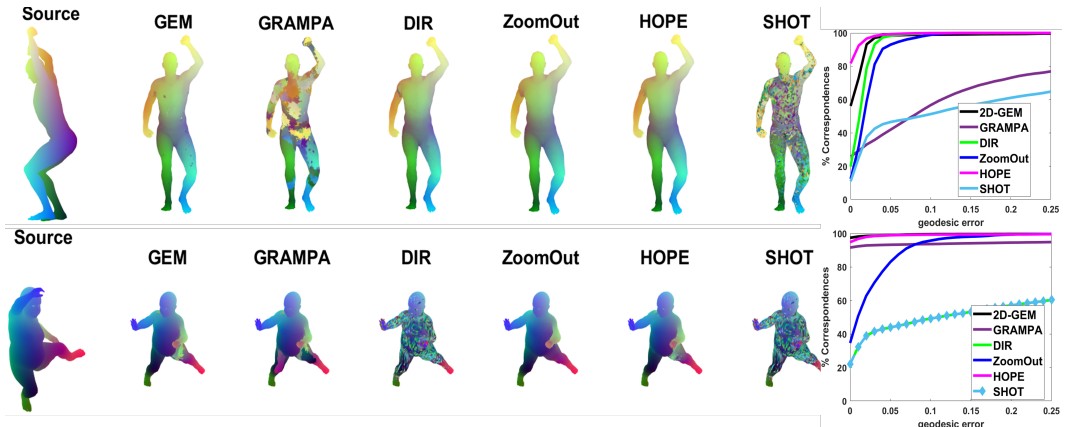

Figure 2: Showing mating of sample shapes from the isometric SCAPE dataset (first row) and the non-isometric TOPKIDS (second row), this figure shows the performance of different baselines.

where $\mathcal{V}$ is the diagonal mass matrix with $\mathcal{V}(i,i)$ entry along the diagonal being the vertex area of vertex $i$, and the matrix $\mathcal{S}$ being defined as:

$$\mathcal{S} = \begin{cases} \frac{1}{2}(cot\alpha_{i,j} + cot\beta_{ij}), & j \in neigh(i) \\ -\sum_{k \in neigh(i)} \mathcal{S}_{i,k} & i = k \\ 0, & otherwise \end{cases} \tag{4}$$

where $neigh(i)$ denotes the neighborhood of $i$. It can be observed from the way the LBO is defined that a non-isometric deformation which changes angles, areas, or even merges part of the shape (such as gluing hands to face for example) will change the LBO and by implication its spectrum, hence changing the vertex-wise and pair-wise descriptors based on the LBO. Figure 2 shows the poor performance of DIR [49] and ZoomOut [31] (which use the spectrum of the LBO as vertex-wise descriptors) on a pair of sample shapes from the non-isometric dataset TOPKIDS [26] (which contains shapes with topological noise).

Other methods aim for accuracy use more unique descriptors. However, these descriptors are generally not smooth since they generally do not capture neighborhoods and as such do not match neighborhoods. Figure 1 shows the first SHOT [45] descriptor and figure 2 show that while it generally provides an accurate map [45] (seen by the proportion of vertices at geodesic error 0) the map is not smooth (as seen by the lack of rapid increase of the geodesic curve). That is why methods aiming for smoothness [31, 37, 12, 17] only use these unique descriptors for initialization and then use the LBO or other smooth basis for refinement.

Others have used the 1-hop [14] or 2-hop [52, 22] neighborhoods of vertices as witnesses to improve an initialized map. Though these methods aim for both accuracy (via uniqueness of vertex neighborhoods), and smoothness via using neighborhoods as witnesses to improve the map (matching neighborhoods that agree best), they are nonetheless affected by the fact that 1-hop or 2-hop neighborhoods may not be unique enough and distant vertices may have similar 1-hop and or 2-hop neighborhoods especially in the presence of symmetries in $\mathcal{A}$. For example, consider GRAMPA [14] which initializes the map by using a graph matching kernel in the full graph spectrum and then uses the 1-hop (or 2-hop as in this figure) neighborhood to refine the map. Figure 2 shows GRAMPA [14] performing well on non-isometric shapes from TOPKIDS where there are few symmetries (and other graph properties) which normally cause the neighborhoods of the vertices to be non-unique. However, figure 2 shows that GRAMPA struggles on a pair of shapes from the nearly isometric dataset SCAPE [26] where shapes have symmetries (and other graph properties) that normally cause the neighborhoods of the vertices to be non-unique. To address this challenge, 2D-GEM[22] proposed to use the concept of local map distortion LMD[49, 48] and the spectrum of the uniform shape laplacian. However, though 2D-GEM seems to perform well on non-isometric and isometric shapes (figure 2), 2D-GEM does not generalize, but rather significantly changed their model (depending on user-defined parameters) in order to handle isometric cases as discussed in Section 4.4.

# 4 HOPE

To address both accuracy and smoothness for correlated meshes undergoing isometric and non-isometric deformations we propose HOPE (k-HOP niEghborhood matching). HOPE is based on the iterative refinement of an initialized map by: (1) using the local map distortion (LMD)[49, 48] to identify poorly matched vertices, and (2) using noise robust k-hop neighborhood-based descriptors for refinement of the maps of these poorly matched vertices.

This section is organized as: (a) the local map distortion, (b) k-hop pairwise descriptors, and (c) iterative refinement pipeline and algorithm.

## 4.1 Local Map Distortion (LMD)

Let $\mathcal{T} : \mathcal{M} \rightarrow \mathcal{N}$ be a map between two shapes. The LMD [49, 22] of the map $\mathcal{T}$ at the vertex $x_i$ is given as follows:

$$\mathcal{D}_\gamma(\mathcal{T})(x_i) = \frac{\sum_{x_j \in \mathcal{B}_\gamma(x_i)} \mathcal{V}_\mathcal{M}(j) DE_\mathcal{T}(x_i, x_j)}{\sum_{x_j \in \mathcal{B}_\gamma(x_i)} \mathcal{V}_\mathcal{M}(j)}, \tag{5}$$

where $\mathcal{B}_\gamma(x_i) = \{x_j \in \mathcal{M} \mid d_\mathcal{M}(x_i, x_j) \leq \gamma\}$ is the $\gamma$-geodesic ball of $x_i$, $\mathcal{V}_\mathcal{M}$ is the area element of the mesh of shape $\mathcal{M}$, and $DE_\mathcal{T}(x_i, x_j) = |d_\mathcal{M}(x_i, x_j) - d_\mathcal{N}(\mathcal{T}(x_i), \mathcal{T}(x_j))|/\gamma$ represents a pairwise distance distortion of mapping nearby vertices $x_i$ and $x_j$ to $\mathcal{T}(x_i)$ and $\mathcal{T}(x_j)$. A smaller value of $\mathcal{D}_\gamma(\mathcal{T})(x_i)$ means a better map continuity of $\mathcal{T}$ at the vertex $x_i$, in other words, the local distance at the point $x_i$ is well preserved. Based on the above definition of LMD, one can check that if $\mathcal{T}$ is an isometric map, then $\mathcal{D}_\gamma(\mathcal{T})(x_i) = 0, \forall x_i \in \mathcal{M}, \gamma > 0$. Conversely, if $\mathcal{D}_\gamma(\mathcal{T})(x_i) = 0, \forall x_i \in \mathcal{M}$ for some $\gamma > 0$, then $\mathcal{T}$ is isometric. We use the LMD to find well-matched pairs $(lmks)$ i.e., $lmks = \{(x_i, \mathcal{T}(x_i)) | \mathcal{D}_\gamma(\mathcal{T})(x_i) \leq \epsilon\}$, where $\epsilon$ is a threshold, We fix $\epsilon$ to be the same values for all dataset for all our experiments. We then call the rest of the vertices on shape $\mathcal{M} \notin lmks$ as non-landmarks $Nlmks$ (i.e., poorly mapped vertices).

## 4.2 K-Hop Pairwise Descriptors

Given an initial map $\mathcal{T}^0$ matching a fraction $\beta$ of the vertices $n$ correctly and the fraction $1 - \beta$ incorrectly, it has been shown that one can refine the map by (a) using the 1-hop [29] neighborhood contained in the graph adjacencies of the triangulations $\mathcal{A}_\mathcal{M}$ and $\mathcal{A}_\mathcal{N}$, or (b) using k-hop [52] neighborhoods contained in binary matrices $\mathcal{A}_{\mathcal{M},k}$ and $\mathcal{A}_{\mathcal{N},k} \in \mathbb{R}^{n \times n}$ indicating whether vertices are connected at a path of length $k$ on the shape mesh adjacencies (note that $k = 1$ is the adjacency). The refinement is done by solving:

$$\mathcal{T}^t = \arg\max_\mathcal{T} \mathbf{Tr}(\mathcal{A}_{\mathcal{M},k}(\mathcal{T}, \mathcal{T}^{t-1})\mathcal{A}_{\mathcal{N},k}), \tag{6}$$

the main assumption being that 2 the meshes $\mathcal{M}$ and $\mathcal{N}$ are correlated i.e., their adjacencies are assumed to come from the same parent graph, with their correlation ratio being $s$. Specifically, given a $\mathcal{G}(n, p)$ Erdos Regny parent graph and the correlation $s$, $1 - s$ is the probability of independently randomly deleting edges from this graph to either obtain the adjacency of $\mathcal{M}$ or that of $\mathcal{N}$. For $k = 1$ and $k = 2$, rigorous analysis and tight bounds were given for recovering the ground truth map $\hat{\mathcal{T}}$ (for given values of $s, p$ and $\beta$) by solving equation 6 (see [51, 30, 52, 29] for exact bounds).

**Theorem 4.1** *Given the k-hop based descriptors $\mathcal{A}_{\mathcal{M},k}$ and $\mathcal{A}_{\mathcal{N},k}$, solving equation 6 is matching vertices whose neighborhoods agree best under $\mathcal{T}^{t-1}$.*

**Proof 4.1** *Notice that:*

- *columns of $\mathcal{A}_{\mathcal{M},k}$ and $\mathcal{A}_{\mathcal{N},k}$ are descriptors for the vertices (rows), and these columns are the k-hop neighborhoods of vertices (rows)*

- *$\mathcal{T}^{t-1}$ is used in equation 6 for first aligning these descriptors i.e., $\mathcal{A}_{\mathcal{M},k}(:, \mathcal{T}^{t-1})$ rearranges the columns of $\mathcal{A}_{\mathcal{M},k}$ thus realigning the descriptors of each vertex (the rows),*

- *each entry $\mathcal{K}(i, j)$ in the product $\mathcal{K} = \mathcal{A}_{\mathcal{M},k}(:, \mathcal{T}^{t-1})\mathcal{A}_{\mathcal{N},k}$ will be the number of vertices that are common in the k-hop neighborhoods of vertices $i$ and $j$, after the alignment of the k-hop neighborhoods of vertices in $\mathcal{M}$ by $\mathcal{A}_{\mathcal{M},k}(:, \mathcal{T}^{t-1})$*

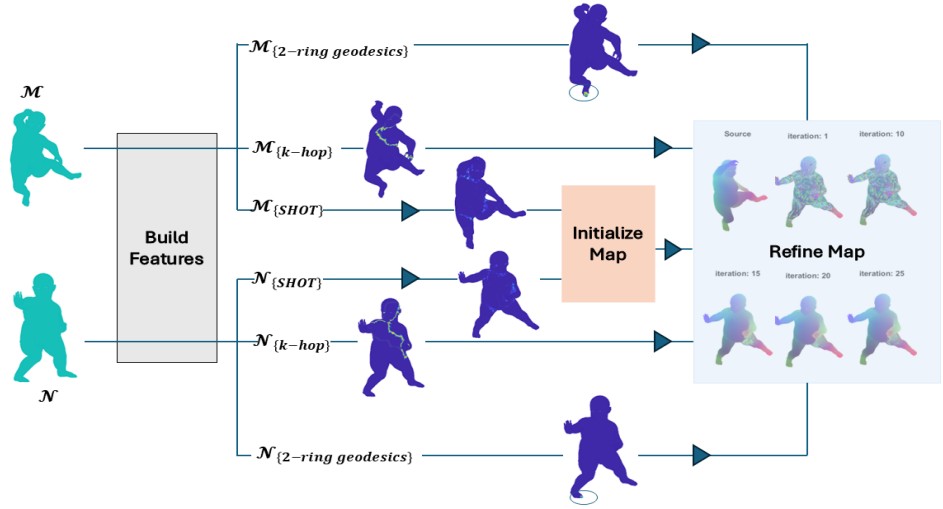

Figure 3: HOPE (a pipeline) for shape matching.

- *finding $\mathcal{T}$ via equation 6 matches the vertices $i$, and $j$ whose k-hop neighborhood have most vertices in common based on the alignment $\mathcal{T}^{t-1}$.*

However, in the presence of symmetries the next 1-hop and 2-hop neighborhoods may not be unique enough as even distant nodes may have the same 1 and or 2-hop connectivity. To address this non-uniqueness, we follow 2D-GEM, and use the LMD to first detect $Nlmks$ vertices. Unlike 2D-GEM which used the laplacian spectrum to refine the map $\mathcal{T}(Nlmks)$ for $Nlmks$ vertices, we instead propose to iteratively use different k-hop neighborhoods for refining the map $\mathcal{T}(Nlmks)$ for $Nlmks$ vertices, i.e., iteratively using nodes at different lengths $k$ to refine $\mathcal{T}(Nmlks)$. We propose this strategy because: (b) first, 2D-GEM's strategy of using the spectrum will suffer from non-uniqueness according to theorm 3.1, and will also not adapt to non-isometric shapes as discussed in Sections 3 and 4.4. (b) Second, according to theorem 4.1 our strategy will ensure that $\mathcal{T}$ is consistent across different neighborhoods. We do so for $1 \leq k \leq k_{max}$ where $k_{max}$ is large. This strategy is inspired by the fact that it was shown that large neighborhood statistics are essential in graph matching [51, 30, 52, 29, 10, 32], and other graph tasks [18, 5, 47, 21, 22, 53].

## 4.3 Pipeline for HOPE

Here we show the general pipeline for HOPE which consists of the two aforementioned steps, namely:

- (a) Map initialization: we use any robust map initializations e.g., the map initialized from SHOT[45] in our experiments
- (b) map refinement for $t$ iteration steps consisting of:
  - Using the LMD [22, 49, 48] to detect the poorly matched $Nlmks$ vertices
  - refine the map for the $Nlmks$ vertices via enforcing different noise robust k-hop neighborhoods consistency (for $1 \leq k \leq k_{max}$) by solving:

$$\mathcal{T}^t(Nlmks) = \arg\max_{\mathcal{T}} \mathbf{Tr}(\mathcal{A}_{\mathcal{M},k}(\mathcal{T}(Nlmks), \mathcal{T}^{t-1})\mathcal{A}_{\mathcal{N},k}), \quad (7)$$

- (c) returning the final map $\mathcal{T}^t$

## 4.4 HOPE vs 2D-GEM

In this section, we briefly introduce 2D-GEM [22] and highlight some key differences between 2D-GEM and HOPE. Like us, 2D-GEM initializes its map via SHOT[45]

and then at each iteration $t$ 2D-GEM refined the map by:

- (1) updating the map $\mathcal{T}$ by solving one iteration of equation 6 using the 2-hop neighborhoods, meaning $k = 2$ in equation 6,

- (2) finding the $lmks$ well-matched pairs, and poorly matched pairs $Nlmks$ pairs using the LMD.

- (3) using the $lmks$ vertices to update the map of the $Nlmks$ vertices in the spectrum by using the GMWM [52] on a cost matrix built from the spectrum of the laplacian of the two shapes (see their paper for more details). They showed similarity between this approach and the functional map [34] approach.

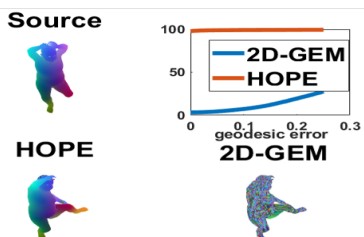

Figure 4: Comparison on TOPKIDS showing HOPE using $\epsilon$ as in Section 5.5 for its LMD threshold (same as in the isometric cases), against 2D-GEM with $\epsilon$ set as in the isometric case (i.e., using the isometric parameters in [22] rather than $\epsilon = 100$ as in the non-isometric case).

The main two differences between 2D-GEM and HOPE are: (a) HOPE completely removes their step (1), and (b) they used the laplacian (LBO) spectrum via their proposed 2D-graph convolution to refine the map for the $Nlmks$ in step (3). However, as mentioned in section 3 and shown in figure 1, using LBO descriptors loses uniqueness since vertices in similar clusters will be grouped together, and in addition, these descriptors are not robust to non-isometric deformations as discussed in Section 3.

A consequence of this is that, 2D-GEM[22] needed to deactivate step (2) and (3) of their algorithms (the $Nlmks$ refinement step) by setting $\epsilon = 100$ as their LMD threshold when dealing with non-isometric shapes. Hence though effective, their algorithm can be seen as two disjoint algorithms that are used for isometric or non-isometric shapes based on the setting of $\epsilon$. But in real life, in some cases, it is not obvious whether the deformation used is isometric or not. On the other hand, we propose refining the $Nlmks$ map based on the preservation of different k-hop connectivity which is a constraint that holds for isometric and non-isometric shapes with correlated triangulations [51, 30, 52].

Figure 4 shows a comparison between HOPE and 2D-GEM on a non-isometric pair of shapes from TOPKIDS using the 2D-GEM $\epsilon$ parameters proposed for isometric shapes. It can be seen that when we use the isometric parameters of 2D-GEM on non-isometric shapes, 2D-GEM fails to perform well (the same holds when using their non-isometric parameter $\epsilon = 100$ on isometric shapes.

### 4.5 Time complexity analysis

See the algorithm for HOPE in Appendix A. Given that for step (1) HOPE follows DIR[49] and [22] in using LMD, checking the LMD takes $O(n)$. The GMWM used to solve step (2) of HOPE takes $O(|Nlmks|^2 logn)$. Hence the total time complexity of HOPE for all $t$ iterations is $O(t(|Nlmks|^2 logn + n))$.

## 5 Experiments

We report experimental results that validate the effectiveness, efficiency, and generalization ability of HOPE in the matching of nearly-isometric and non-isometric 3D shapes.

### 5.1 Experimental Set-up

All experiments are conducted in Matlab 2023 on a Windows 11 system with 32GB RAM and Intel(R) i5 13500 CPU @ 2.50-4.8GHz.

### 5.2 Datasets

We evaluate the performance of HOPE on two nearly isometric benchmark datasets TOSCA [7], and SCAPE [3], as well as on the non-isometric dataset SHREC'16 (TOPKIDS) [26], TOPKIDS

contains 25 shapes of the same class with up to 12K vertices, undergoing near-isometric deformations in addition to large topological noise (such as merging hands to thighs) which results in $n_{\mathcal{M}} \neq n_{\mathcal{N}}$. TOSCA consists of 80 shapes in 8 different categories (human and animal shapes) with vertex numbers ranging from 4k to 50k. SCAPE has 71 shapes (12,500 vertices for each) of the same person with different poses.

Furthermore, to see the generalization abilities of HOPE, we equally used datasets SCAPE_r[9], FAUST_r[9], and TOSCA_r[9], which are remeshed shape datasets. The SCAPE_r consists of the same 71 shapes from the SCAPE dataset, but remeshed, while the FAUST_r (TOSCA_r) likewise contains the same FAUST (TOSCA) datasets but remeshed. We used the same 71 test pairs SCAPE_r as for SCAPE, and for FAUST_r and TOSCA_r we followed [9]

### 5.3 Evaluation Metrics

We use the geodesic error as our error metric [14, 22]. Given that the map of an algorithm maps $x_i \in \mathcal{M}$ to $x_j \in \mathcal{N}$, and the true map maps $x_i$ to $x_j^*$, the geodesic error is defined as $e(x_i) = \frac{d_{\mathcal{N}}(x_j, x_j^*)}{diam(\mathcal{N})}$, where $d_{\mathcal{N}}$ denotes the geodesic distance on $\mathcal{N}$, and $diam(\mathcal{N})$ is the geodesic diameter of $\mathcal{N}$.

### 5.4 Baselines

Following 2D-GEM [22], we compare HOPE with the following methods: EM [42], GE [25], RF [40], PFM [39], FSPM [28], Kernel-Matching [26], and GRAMPA [14], [22], SGMDS[1], FM[34], BIM[24], Mobius[27], Best-Conformal [24], Kernel-Matching [26], DIR-500 [49] which uses 500 eigenvectors, DIR-1000 [49] which uses 1000 eigenvectors.

### 5.5 Parameter Settings

**On ZoomOut**, we start with a functional map using the first 20 eigenvectors of the LBO, then we iteratively add an eigenvector until we reach the 120th eigenvector after which we stop.

**On all other Baselines**, we follow the settings from [22].

**On HOPE**, on all datasets we set the LMD threshold $\epsilon$, staring from $\epsilon = 100$ and 10 equally spaced values to $\epsilon = 0.2$ i.e., we use $\epsilon = linespace(100, 0.2, 10)$ and we set $t = 60$. When the last value of $e$ is reached, it is maintained for the rest of the iterations. We equaly set $k_{max} = 8$ for all datasets. For the LMD, we used the second ring neighborhood following [49, 22].

### 5.6 Performance Analysis

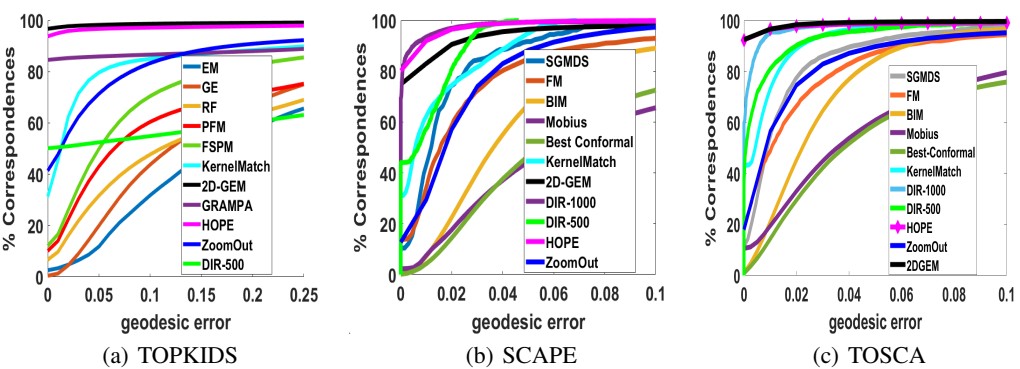

(a) TOPKIDS      (b) SCAPE      (c) TOSCA

Figure 5: Performance comparison on TOPKIDS 5(a), SCAPE 5(b), and TOSCA 5(c).

**Comparison on non-isometric shapes.** On the dataset with topological noise TOPKIDS (Figure 5(a)), the top 3 methods are the noise-robust methods 2D-GEM with $96.7\%$ vertices correctly matched, followed by HOPE with $94.9\%$, and GRAMPA with accuracy of $84.5\%$. This shows that 2D-GEM, HOPE, and GRAMPA are indeed adapted for non-isometric shapes. Moreover, as discussed in

[22], GRAMPA is very time-consuming as it needs the full basis of the graph adjacencies of the shapes, while HOPE and 2D-GEM do not. For example on a shape with 12500 vertices from SCAPE, GRAMPA takes roughly 118 seconds on our set up while 2D-GEM and HOPE take 56 and 52 seconds respectively. Better still HOPE is a general model as it can be seen that with the same parameter of $\epsilon$ (see Section 5.5) for its LMD, it performs well on all datasets.

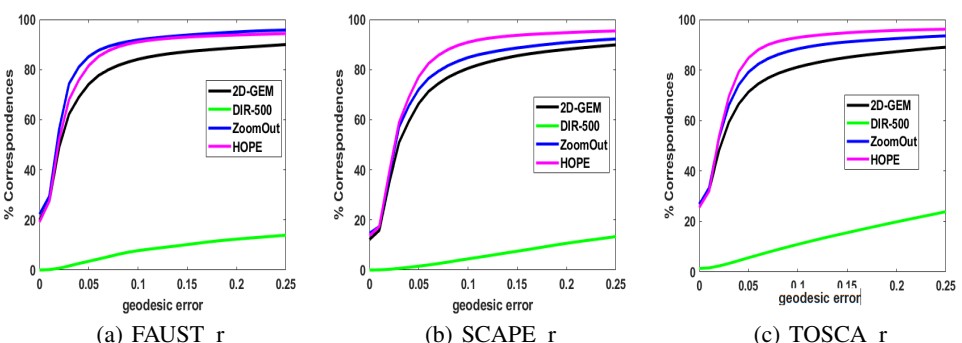

| (a) FAUST_r | (b) SCAPE_r | (c) TOSCA_r |

Figure 6: Performance comparison on FAUST_r 6(a), SCAPE_r 6(b), and TOSCA_r 6(c).

**Comparison on isometric shapes**. On isometric shapes, methods that enforce stronger geometric constraints on their refinement pipelines such as 2D-GEM (with appropriate parameters), DIR, and HOPE. HOPE outperform all other baselines. 2D-GEM achieves $92.54\%$ accuracy at geodesic error 0 on the TOSCA dataset, as well as $74.8\%$ accuracy on the SCAPE dataset, while HOPE achieves $80.11\%$ on SCAPE and $92.54\%$ on TOSCA using no matrix decomposition. Third is DIR-1000[49] with 1000 eigenvectors which achieves around $69.5\%$ accuracy at geodesic error 0 on SCAPE, and $59.8\%$ accuracy at geodesic error 0 on TOSCA. Moreover, unlike DIR (and 2D-GEM,), HOPE generalizes to this isometric setting using the same parameters (see Section 5.5).

**Comparison on remeshed shapes**. On the remeshed datasets FAUST_r, SCAPE_r and TOSCA_r in figure 6, methods that enforce stronger geometric constraints such as DIR performed poorly, while those that are more robust to noise such as 2D-GEM (with appropriate parameters), ZoomOut and HOPE perform relatively well. Though 2D-GEM does not outperform ZoomOut in this setting (probably due to differences in the mesh connectivity), HOPE still outperforms ZoomOut though by a smaller margin than on datasets with similar mesh connectivity amongst pairs.

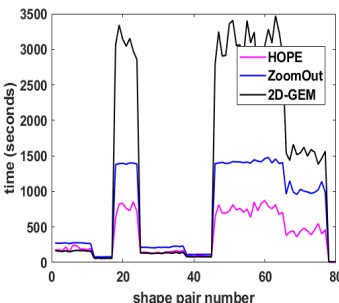

Figure 7: Time Comparison on TOSCA showing HOPE, 2D-GEM, and ZoomOut.

**Time Comparison**. Here we compare the time usage per shape for both 2D-GEM, HOPE, and ZoomOut on the TOSCA dataset. It can be seen from figure 7 that HOPE is relatively faster than 2D-GEM, and even ZoomoOut (for the given parameters of ZoomOut that we used for our experiments).

## 6 Limitations

The main limitation of this work is the reliance on correlated triangulations between the pair of shapes to be matched as outlined in Section 4.2. One can see this drawback on figures 6 which show

that though HOPE still performs well on remeshed shapes (shapes where the triangulations are not very correlated), and even outperforms other baselines, it nonetheless does not perform as well as on figures 5(b), 5(a) and 5(c) where the triangulations of the pairs matched are strongly correlated. This is a major drawback because triangulations are often hard and expensive to get, especially consistent triangulations between shapes.

# 7    Conclusion

We introduced an effective and easy-to-implement map refinement strategy consisting of; (a) detecting poorly matched vertices (nodes) using the concept of local map distortion (LMD), and (b) improving the map of these poorly matched vertices via noise robust k-hop pairwise descriptors. We then conducted a series of experiments to show that our framework is effective and generalizable to different shape datasets. We also discussed the main limitation of our work (the fact that it is reliant on the triangulation consistency between the shapes matched).

# 8    Broader Impact

This work proposes a map refinement strategy for shape matching that is based on matching different k-hop neighborhoods of vertices. It then validates the effectiveness of this strategy on several shape matching datasets. This can spark new research on the importance of large neighborhood statistics for shape matching and other related tasks. As we focus solely proposing a framework for map refinement for shape matching, we do not see clear negative impact of this work.

# 9    Acknowledgements

This work is partially supported by Hong Kong Research Grants Council under the Areas of Excellence Scheme grant AoE/P-601/23-N.

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

# EHOPE: Shape Matching Via Aligning Different K-hop Neighbourhoods
# Appendices:

## A   HOPE Algorithm

In this section, we present the algorithm for HOPE. This is given in algorthm 1

---
**Algorithm 1** : HOPE

---
0. **input** An initial map $\mathcal{T}^0$, distance matrices $d_{\mathcal{M}}$ and $d_{\mathcal{N}}$, LMD threshold $\epsilon$, maximum iteration $t$, and the maximum hop $k_{max}$ and $k = 1$
**while** $0 \leq i \leq t$ **do**
   1. build $\mathcal{A}_{\mathcal{M},k}$ and $\mathcal{A}_{\mathcal{M},k}$
   2. set $k = k + 1$
   **if** $k > k_{max}$ **then**
      3. $k = 1$,
   **end if**
   4. Given $\mathcal{T}^i$, use the LMD to locate well-matched points $lmks$ and poorly-matched points $Nlmks$,
   5. Use these $Nlmks$ pairs to update the $\mathcal{T}^i(Nlmks)$ by using the GMWM to solve equation (7),
**end while**
**return** $\mathcal{T}^t$.

---

## B   Ablation and Parameter Sensitivity Studies

In this section, we conduct parameter and ablation studies. We use the following settings:

- HOPE: the hope algorithm as in algorithm 1, with $t = 60$ and $\epsilon = linespace(100, 0.2, 10)$ as in the experiments in Section 5 in the main paper,
- HOPE-M: where we reduce the number of iterations to $t = 20$ in algorithm 1,
- HOPE-th: where we set $\epsilon = linespace(1, 0.2, 10)$ in algorithm 1,
- HOPE-fixhop: where we simply solve equation 6 with $k = 1$ and $k = 2$ alternatively per iteration as in algorithm 1,
- HOPE-LMDvaryhop: where use $k_{max} = 2$ in algorithm 1,
- HOPE-varyhop: where we simply solve equation 6 with $k = [1, 2, \cdots, 8]$ alternatively per iteration, as in algorithm 1.

It can be observed from figure 8 that the best model overall is HOPE-th which is best on isometric (figure 8(b)) shapes and non-isometric shapes (figure 8(a)) and performaing comparatively to other variants on remeshed shapes (figure 8(c)). This is followed by HOPE. This indicates that the $\epsilon$ we used for our main experiments in Section 5 is sub-optimal since we randomly selected the range to be $linespace(100, 0.2, 10)$ without any parameter tuning to demonstrate the effectiveness and generalizability of HOPE. We notice that the variants without the LMD (HOPE-varyhop and HOPE-fixhop) all struggled on the isometric shape (figure 8(b)) validating our observation that the k-hop neighborhood of nodes may not be very unique especially when there are symmetries, and as such using a stronger constraint like the LMD is beneficial in such settings (Section 3 and 4)

## C   Different Initializations

In this section, we conduct studies on the effects of different initializations on HOPE. We use the following settings:

- HOPE-SHOT: where we use SHOT[45] descriptors to initialize $t^0$ in algorithm 1,
- HOPE-HKS: where we use Heat Kernel Signatures (HKS)[8] descriptors to initialize $t^0$ in algorithm 1,
- HOPE-WKS: where we use Wave Kernel Signatures (WKS)[4] descriptors to initialize $t^0$ in algorithm 1.

Figure 9 shows that indeed SHOT [45] which is commonly used in practice is a robust and good descriptor to use as initialization. On the other hand while both the HKS [8] and WKS [4] provided an initialization that could be enhanced by HOPE relatively well on the isometric shape (figure 9(b)), only WKS and SHOT were suitable for the remeshed shape (figure 9(c)), while only SHOT initializations where relatively okay in terms of accuracy on the non-sometric shape (figure 9(a)).

## D    Partial Shape Matching

In this section, we show the performance of HOPE on partial shape matching where we match the full shapes to the partial shapes. Here we used the SHREC16 HOLES and SHREC16 CUTS following Cao et al. [9]. Figure 10 shows that although HOPE was not designed specifically for the partial shape setting, it nonetheless performs relatively well on average as seen by the average geodesic curve.

## E    Non-Isometric Shape Matching

In this section, we show the performance of HOPE on another non-isometric shape dataset SMAL_r Cao et al. [9], where we perform intra-class matching of 298 pairs of shapes. Figure 11 shows that HOPE again outperforms other baselines even when the mesh triangulations are not strongly correlated (due to noise).

## F    Visual Comparisons

In this section, we show the visual comparisons between HOPE and other baselines, on SCAPE_r (figure 12), SHREC16 (figure 13), and TOPKIDS (figure 14).

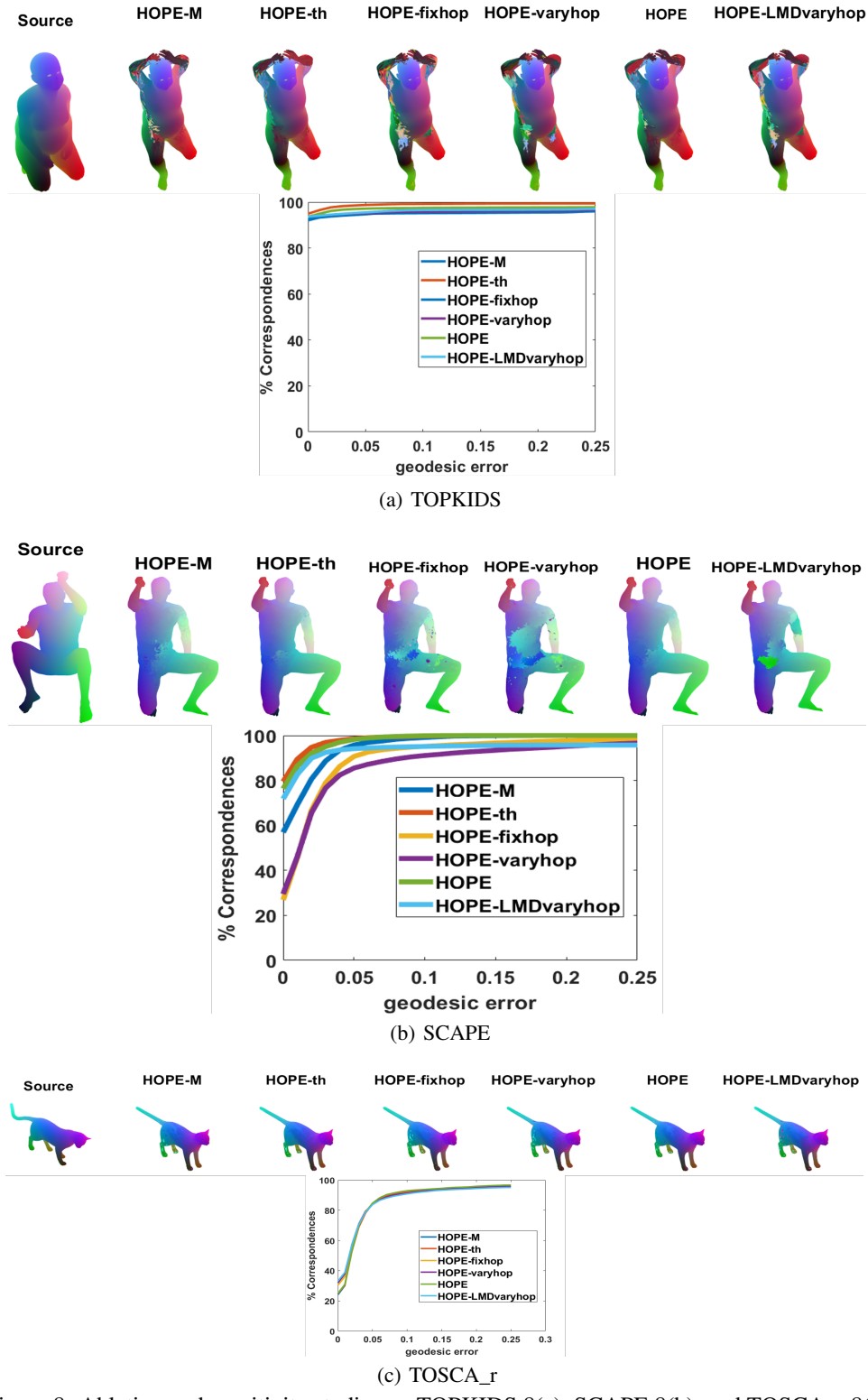

(a) TOPKIDS

(b) SCAPE

(c) TOSCA_r

Figure 8: Ablation and sensitivity studies on TOPKIDS 8(a), SCAPE 8(b), and TOSCA_r 8(c).

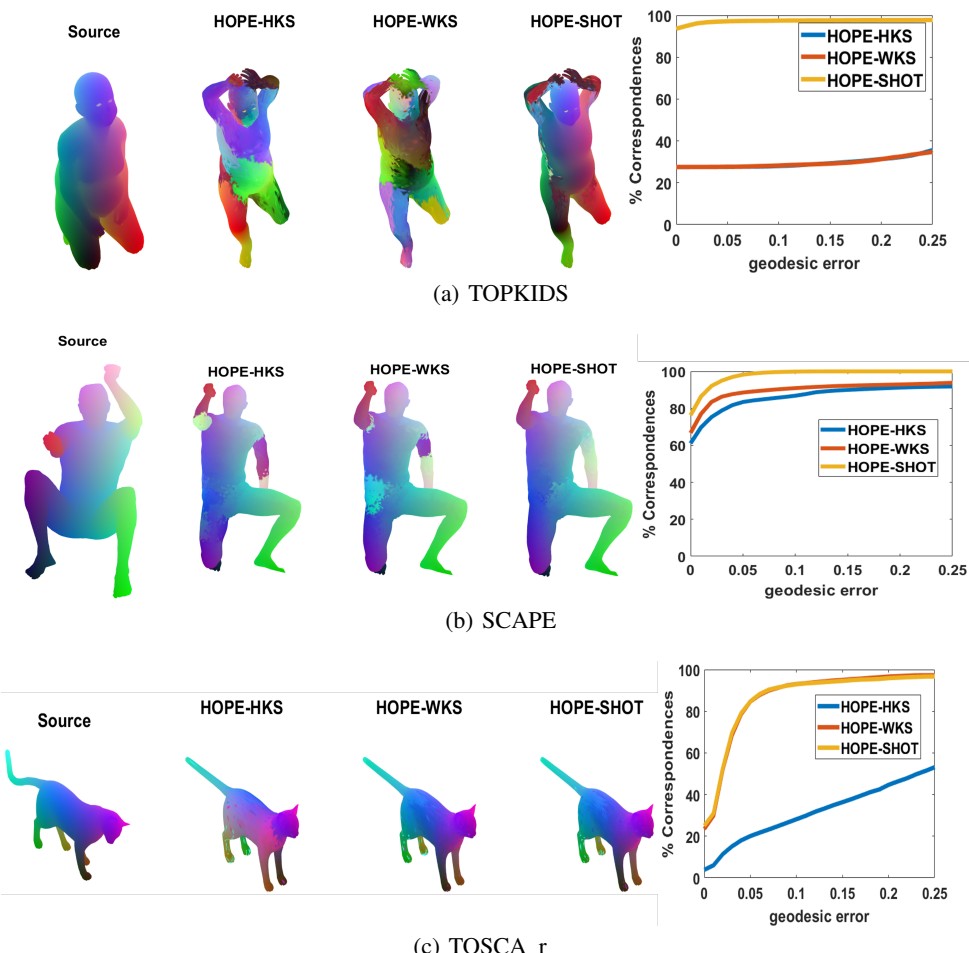

(a) TOPKIDS

(b) SCAPE

(c) TOSCA_r

Figure 9: Different initialzations on TOPKIDS 9(a), SCAPE 9(b), and TOSCA_r 9(c).

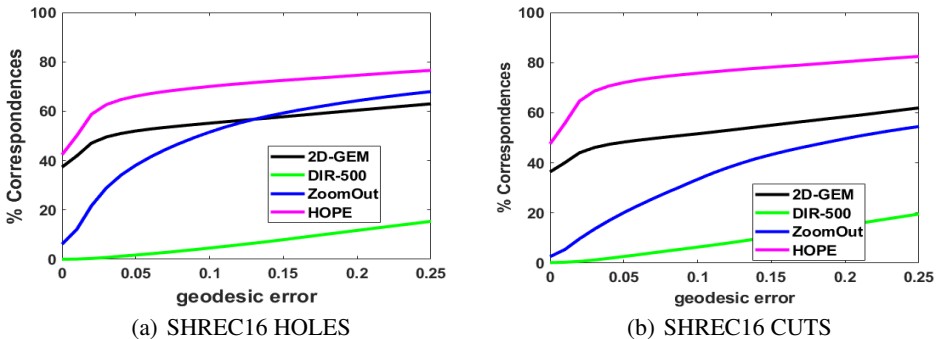

(a) SHREC16 HOLES

(b) SHREC16 CUTS

Figure 10: Performance comparisons on SHREC16 HOLES 10(a), SHREC16 CUTS 9(b), and TOSCA_r 10(b).

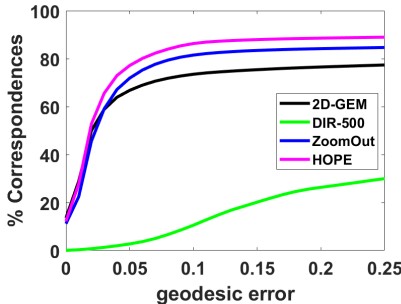

Figure 11: Comparison on intra-class matching on SMAL_r using 298 pairs of shape. Figure showing HOPE, 2D-GEM, ZoomOut, and DIR.

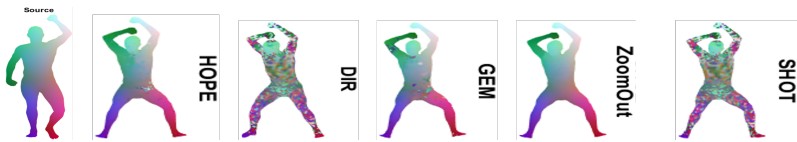

Figure 12: Sample shape from SCAPE_r. Figure showing HOPE, 2D-GEM, ZoomOut, and DIR.

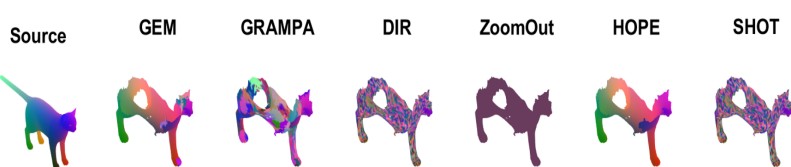

Figure 13: Sample shape from SHREC16-cuts. Figure showing HOPE, 2D-GEM, ZoomOut, and DIR.

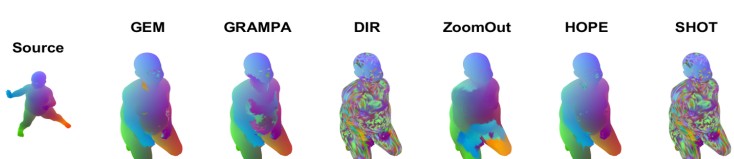

Figure 14: Sample shape from TOPKIDS. Figure showing HOPE, 2D-GEM, ZoomOut, and DIR.

