# OpenReview forum: "HOPE: Shape Matching Via Aligning Different K-hop Neighbourhoods"
_NeurIPS.cc/2024/Conference — NeurIPS 2024 poster_

### Official Review · Reviewer_hBzn · 2024-06-27

**Soundness:** 3
**Presentation:** 4
**Contribution:** 2
**Rating:** 6
**Confidence:** 5

**Summary:**

The paper presents a new method for shape correspondence that extracts descriptors that are both smooth over the manifold and distinctive enough to find high precision point matching. One can view the presented method as an improvement over the 2D-GEM method, that, as opposed to the latter, does not use the eigenfunction of the laplacian and interacts with more the 2-hop neighborhood.  It is extensively tested on various benchmarks in the field, near-isometric and non-isometric ones, and presented promising results.

**Strengths:**

It presented a descriptor that shows improvement over previous ones. The method is well explained. The related work section is written well, and it adds an important context to the paper. Additionally it extensively tested and presented superior results in comparison to previous methods.

**Weaknesses:**

The new method to some extent resembles the 2D-GEM method, which limits the novelty. As presented in the limitation, since it uses the vertex neighborhood it is vulnerable to remeshing, and it counters the statements in the paper about the method usefulness in real data scenarios (overclaiming).
Small issues: I think [1] should be referred to because of its similarly to the ideas presented in the paper, they also use alignment processing of the initial descriptors by reducing geodesic distance error.

[1] Bracha, A., et al., 2020. Shape correspondence by aligning scale-invariant LBO eigenfunctions. 3DOR: Eurographics Workshop on 3D Object Retrieval

**Questions:**

see weaknesses

**Limitations:**

see weaknesses

---

> ### Author Rebuttal · Authors · 2024-07-31
>
> We thank the Reviewer for the insightful and constructive comments. We address some of the Reviewer’s comments and suggestions below:
>
> # Weaknesses:
> - W1: The new method to some extent resembles the 2D-GEM method, which limits the novelty. As presented in the limitation, since it uses the vertex neighborhood it is vulnerable to remeshing, and it counters the statements in the paper about the method usefulness in real data scenarios (overclaiming). Small issues: I think [1] should be referred to because of its similarly to the ideas presented in the paper, they also use alignment processing of the initial descriptors by reducing geodesic distance error.
> [1] Bracha, A., et al., 2020. Shape correspondence by aligning scale-invariant LBO eigenfunctions. 3DOR: Eurographics Workshop on 3D Object Retrieval We thank the reviewer for pointing us to this insightful work.
> - RW1:  We will add this work to our related work section, since indeed [1] also proposed to address some concerns with the LBO basis for shape matching. Specifically, [1] proposed to solve the problem caused by rotation, reflections, sign flips and other symmetries in the LBO basis by first aligning the eigenfunctions (vectors) of the LBO of two nearly isometric shapes and then using these aligned eigen-vectors as the new basis for the shape matching problem.  We also agree with the Reviewer that the limitation incurred by HOPE from using the mesh of the shape is a real one since in the real world most data is meshed independently of each other and so the meshes may not be correlated. We are trying at the moment to study this hard problem and considering how to lift this limitation in a near future work, and will welcome any suggestions from the Reviewer. However, to appease any other concerns from the Reviewer with regards to this, we recall with the Reviewer that though HOPE suffers from this limitation, our experiments in Section 5.6 specifically lines 305-310 as well as figure~5 both help to show that though HOPE does not drastically outperform other baselines on re-meshed shapes, it is still competitive by either performing comparatively or better. Moreover, we attached some more experiments on the SMAL_r (challenging dataset, see PDF attached) as well as some qualitative results on different datasets, which both show again the benefit of choosing HOPE over competitors such as 2D-GEM, DIR, or ZoomOut.

---

> > ### Comment · Reviewer_hBzn · 2024-08-11
> > **Nice paper.**
> >
> > The authors answered our equations, and present a nice methodology to overcome difficulties in shape matching.

---

### Official Review · Reviewer_NAeg · 2024-07-09

**Soundness:** 2
**Presentation:** 2
**Contribution:** 2
**Rating:** 3
**Confidence:** 4

**Summary:**

This paper introduces HOPE, a method that leverages k-hop neighborhoods as pairwise descriptors to achieve both accuracy and smoothness in shape matching. This approach aims to overcome the limitations of existing methods that often struggle to balance these two crucial aspects. The k-hop neighborhood concept involves considering the vertices within a certain distance (k hops) from a given vertex, providing a local context that can be used to compute pairwise descriptors. The authors validate their approach using several benchmark datasets, including SCAPE, TOSCA, and TOPKIDS.

**Strengths:**

1. The paper is well-structured, with a clear organization.
2. The method section provides a theoretical argument result. The theoretical results demonstrate the validity of the method.
3. The authors provide a comparative explanation with the main baseline, 2D-GEM.
4. The method achieves good results in partial shape matching.

**Weaknesses:**

1. Novelty: The methods in the paper lack overall novelty, as the proposed approach involves minor optimizations and modifications based on existing work.
2. Parameter Sensitivity: Although not explicitly discussed in the provided sections, methods that involve iterative refinement and multiple descriptors often require careful tuning of parameters. The performance of HOPE may depend on the selection of parameters such as the number of hops (k) and the weights used in the LMD optimization. A thorough analysis of the method's sensitivity to different parameters would be beneficial, possibly including guidelines or heuristics for parameter selection.
3. Experimental Results: In the compared datasets, this method does not show a significant improvement over the baseline. The advantages of the technique are not evident from the provided qualitative and quantitative results. The paper mentions other works that address smoothness and accuracy but could benefit from a more detailed comparative analysis, showing direct performance metrics against the most recent and relevant methods.
4. Benchmark: There are relatively few results on non-isometric datasets in the comparison. It is suggested to supplement results from other datasets, such as SMAL_r [a], DT4D-H [b], or other data in SHREC07 [c].
[a] Deep Orientation-Aware Functional Maps: Tackling Symmetry Issues in Shape Matching.
[b] Smooth Non-Rigid Shape Matching via Effective Dirichlet Energy Optimization.
[c] http://partial.ge.imati.cnr.it.

**Questions:**

1. What is the impact of the initialization map quality on this method, and how does it compare in tolerance level with other baselines?
2. How much does the method improve over the baseline in specific quantitative results on the SHREC16 dataset? There are no qualitative results or specific quantitative results, including geodesic error.
3. There are no quantitative metrics provided to demonstrate the smoothness of the map.
4. What is the value of kmax set in the experiments, and is it consistently optimal across different datasets?

**Limitations:**

Please refer to the Weaknesses and Questions above.

---

> ### Author Rebuttal · Authors · 2024-07-31
>
> We thank the Reviewer for the constructive and insightful comments. We will strive to the best of our ability to address all concerns in hopes that the Reviewer can please reconcider the reviews .
>
> # Weaknesses:
> - W1: Novelty: The methods in the paper lack overall novelty, as the proposed approach involves minor optimizations and modifications based on existing work.
> - RW1: To the best of our knowledge, we are the first and only work that use different k-hop neighborhoods as pairwise descriptors to refine the map in shape matching.
> - W2: Parameter Sensitivity: Although not explicitly discussed in the provided sections, methods that involve iterative refinement and multiple descriptors often require careful tuning of parameters. The performance of HOPE may depend on the selection of parameters such as the number of hops (k) and the weights used in the LMD optimization. A thorough analysis of the method's sensitivity to different parameters would be beneficial, possibly including guidelines or heuristics for parameter selection.
> - RW2: We please refer the Reviewer to lines 163-164 and lines 284-286 where we mention that all our parameters are fixed for all datasets and to appendix B were we conduct the parameter sensitivity which show that HOPE is not sensitive to parameters chosen.
> - W3: Experimental Results: In the compared datasets, this method does not show a significant improvement over the baseline. The advantages of the technique are not evident from the provided qualitative and quantitative results. The paper mentions other works that address smoothness and accuracy but could benefit from a more detailed comparative analysis, showing direct performance metrics against the most recent and relevant methods.
> - RW3: We please recall with the Reviewer that as shown in Section 5.6 and in appendix B, HOPE performs significantly better than all baselines on 8 different datasets while using the same set of parameters with no need of tweaking.
> - W4: Benchmark: There are relatively few results on non-isometric datasets in the comparison. It is suggested to supplement results from other datasets, such as SMAL_r [a], DT4D-H [b], or other data in SHREC07 [c]. [a] Deep Orientation-Aware Functional Maps: Tackling Symmetry Issues in Shape Matching. [b] Smooth Non-Rigid Shape Matching via Effective Dirichlet Energy Optimization. [c] http://partial.ge.imati.cnr.it.
> - RW4: We will add and briefly expound the suggested works in the related work section. We conducted interclass matching on SMAL_r as suggested givin a total of 298 pairs averaged on the curve. Please see the results in the attached PDF.  Moreover,  we also believe that we have provided a wide range of experiments on several datasets to illustrate our point. We plead with the reviewer to look at the variety of datasets we used (up to 8 different datasets, 6 in the main paper in Section 5, and 2 more in Appendix D). So we plead with the reviewer to Review the scores again as we believe our work can indeed be very beneficial to the research community. Finally, for any other tests, we please refer the reviewer to our open-source code provided to test any custom datasets in mind (we will appreciate feedback on how our model performs on those as well).
>
> # Questions:
> - Q1: What is the impact of the initialization map quality on this method, and how does it compare in tolerance level with other baselines? We please refer the Reviewer to Appendices B, C where we discuss in detail with more experiments the parameter sensitivity, ablation studies, and different initializations. We equally please refer the reviewer to Section 5 and Appendix D where on all 6 datasets (in Section 5) and 2 datasets (in Appendix D) with the same parameters HOPE outperforms all other baselines.
> - Q2: How much does the method improve over the baseline in specific quantitative results on the SHREC16 dataset? There are no qualitative results or specific quantitative results, including geodesic error.
> - R2: For the Shrec16 dataset in Appendix D we provided the %correspondences with geodesic error curves on page 17. We apologize that it is on this page. We will move this to a better location.
> - Q3: There are no quantitative metrics provided to demonstrate the smoothness of the map.
> - RQ3: Though we did not use an explicit metric for the smoothness, as noted in lines 50-53, one can infer this by observing the rapid increase of the %Correspondence as it moves away from geodesic error 0 on the plots. We will welcome any smoothness metric suggested by the reviewer and add it to our work, since the ones we are familiar with such as conformal distortion, bijectivity, chamfer distance and so on all do not measure smoothness explicitly. Hence we resorted to observing how fast 100% Correspondence accuracy is reached as one moves away from geodesic error 0 closest one to the best of our knowledge.
> - Q5: What is the value of kmax set in the experiments, and is it consistently optimal across different datasets? Kmax is fixed to 8 for all experiments in the paper.
> - RQ5: We apologize to the Reviewer for not making this explicit in lines 284-286 (where we mention that all our parameters are fixed for all datasets). We will add this in the final version of the paper.

---

> > ### Comment · Reviewer_NAeg · 2024-08-11
> >
> > Thanks to the authors for their thorough response and the additional experiments on SMAL_r. I carefully reviewed the authors' replies and the corresponding sections of the paper. However, I believe that the response does not address my main concerns.
> >
> > First, I think the paper lacks novelty. The authors emphasize using different k-hop neighborhoods, but I believe this is merely an extension of previous work without introducing new insights. Additionally, the theoretical proof provided regarding k-hop is confusing.
> >
> > Regarding the experiments, specifically Fig. 4, I do not believe the results significantly improved over the baseline. While there seems to be some improvement for remeshed meshes, I suggest discussing the reasons behind these results. Moreover, I do not observe a significant improvement over GEM in most cases in the qualitative examples chosen in both the main text and supplementary materials.
> >
> > Another suggestion is to provide the mean geodesic error values along with the curves so that the improvement of the method can be directly compared. I believe the entire work requires significant improvements in both writing and innovation, and I do not think this work is ready for publication in NeurIPS 2024.

---

> ### Author Response · Authors · 2024-08-11
> **Response 2 to  Reviewer NAeg**
>
> *NOVELTY*
>
> Please find below the novelty of HOPE (our work), especially when compared to GEM:
>
> -	First, our proposed HOPE is the first work that uses different K-hops neighborhoods as witnessed for iterative map refinement for shape matching. We justified this in Section 4.2 (supported by theorem 4.1 )
> -	Second, we showed in section 4.4 (using theorems 3.1 and 4.1) that GEM does not generalize to different datasets without a significant change in the algorithm. We showed that changing the parameters as they did in their paper is really changing the algorithm, and as such GEM is really two different algorithms from which we choose one depending on the parameters, because choosing large thresholding parameters as they do for non-isometric shapes, re-meshed shapes, and partial shapes deactivates the LMD. This is a huge disadvantage because generally in real life one does not know whether the shapes one is dealing with are isometric or not.
>
> *THEORETICAL PROOF*
>
> We would like the Reviewer to highlight which part of the proof of the usefulness of the k-hop neighborhood was unclear so that we can address this. Since the Reviewer previously did not mention anything about this, else we would have done our best to address this.
>
> *EXPERIMENTAL PERFORMANCE*
>
> With regards to the experiments, the only baseline that looks competitive is indeed GEM, but again:
>
> -	HOPE outperforms GEM in most datasets. GEM does not generalize. The only reason GEM even performs well in our experiments in Section 5 and the Appendix is because we significantly changed their algorithm by using different set of parameters for different datasets as they did in their paper (since choosing large thresholding parameters as they do for non-isometric shapes, re-meshed shapes, and partial shapes deactivates the LMD).  We showed this in Section 4.4 where we used a toy example from the TOPKIDS dataset to prove that if we do not change the parameters, GEM will perform extremely poorly since GEM is really two algorithms from which using different sets of parameters will select one.
> -	So really HOPE generalizes to all Datasets while outperforming GEM significantly in the re-meshed (Figure 5) and partial shapes (Figure 9) as well as the SCAPE (Figure 4). Moreover, in addition to outperforming GEM, HOPE also outperforms all other baselines in all experiments (by a large margin in some cases).
> - We discussed in Section 4 that the Reason HOPE can generalize are: (a) first that the K-hops are noise robust (thus addressing the re-meshed shapes and noises to some extent), and (b) secondly are more unique than other shape descriptors.
>
> *REPORTING MEAN GEODESIC ERROR*
>
> With regards to reporting the mean geodesic errors:
>
> - In case of acceptance, just like the SMAL_r experiments we added, we can also add these to our Appendix as suggested.

---

### Official Review · Reviewer_E6Vo · 2024-07-11

**Soundness:** 1
**Presentation:** 1
**Contribution:** 1
**Rating:** 2
**Confidence:** 5

**Summary:**

This submission proposes a descriptor utilizing k-hop neighborhoods for non-rigid 3D shape matching. The descriptor is used jointly with local map distortion for map refinement.

Overall, I am highly frustrated during reviewing this submission. While I have tried my best to parse and understand the technical details, the confusing formulations and derivations, together with missing definitions have prevented me from doing so. Overall, I would encourage the authors to revise carefully the submission, so that it is readable.

**Strengths:**

To be honest, I have trouble reading out the paper, therefore I can hardly conclude any strength for real.

**Weaknesses:**

The paper lacks readability, especially on the technical part. An incomplete list of frustrations I have encountered is as follows:

1. The presentation is poor. Starting from Eqn.(1), I have got lost. There has never been an explicit formula or at least a clear reference to the prior works for W_M(T^t, T^{t-1}). Similarly Q_M, Q_N are not defined. I have tried my best to parse the statements, but have to give up in the end.

2. Theorem 3.1 and its proof are also confusing. First of all, Q_M and Q_N in proof 3.1 are *not* mentioned in the theorem. Second, functional maps by its definition is defined regarding the basis, instead of some arbitrary vertex-wise descriptor. However, I did not anything around L95 saying Qs are eigenbasis (or any kind of basis regarding C).

3. In Eqn.(6), the definition of A_M, k(T, T^{k-1}) is again undefined.

4. The proof of Theorem 4.1 is pointless. Please use the standard mathematical derivation, rather than piling up a series of unverified statements.

**Questions:**

1. I would disagree with the argument in L27. Shape matching methods would rarely aim for solely smoothness without accuracy -- the latter is the most important, while the former can serve as good regularization.

2. Should niegh(i) in Eqn.(4) be neigh(i)?

3. In Fig.4, I can not see the curve regarding HOPE. It would be useful to report geodesic error in the legend.

4. If I understood correctly, the appendix should be behind checklist.

**Limitations:**

It looks fine to me.

---

> ### Author Rebuttal · Authors · 2024-07-31
>
> We thank the Reviewer for the clarification seeking comments. We sincerely apologize to the reviewer for any confusion or misunderstanding. We will strive to the best of our ability to address all concerns. However, considering the good scores of other Reviewers, we strongly plead with the Reviewer to please reconsider the reviews and to please ask us any further questions that may help in this regard:
>
> # Weaknesses:
> - W1: The presentation is poor. Starting from Eqn.(1), I have got lost. There has never been an explicit formula or at least a clear reference to the prior works for W_M(T^t, T^{t-1}). Similarly Q_M, Q_N are not defined. I have tried my best to parse the statements, but have to give up in the end. W_M(T^t, T^{t-1}), Q_M, and Q_N are all defined in the paragraph below equations 1 and 2. See lines 64-76.
> - RW1:  Equations 1 and 2 are general formulations of the iterative refinement strategy of methods such as Kernel Matching[1], 2D-GEM[2] and GRAMPA[3], and others. For example  the minimization in equation 1 is equivalent to the maximization  argmax_{T^t} Tr(W_M(T^t, T^{t-1}) W_N), where T^t is the map we are trying to obtain at iteration t and the map T^{t-1} is the map that was previously obtained in iteration t-1 (which is done by all 3 papers namely; Kernel Matching , 2D-GEM and GRAMPA for example).
> - W2: Theorem 3.1 and its proof are also confusing. First of all, Q_M and Q_N in proof 3.1 are not mentioned in the theorem. Second, functional maps by its definition is defined regarding the basis, instead of some arbitrary vertex-wise descriptor. However, I did not anything around L95 saying Qs are eigenbasis (or any kind of basis regarding C).
> - RW2: First, we want to agree with the reviewer here that a functional map can be computed in any basis and not only in the eigen basis (see section~4 of the original functional map paper). As such, given any basis (e.g., first k eigen-vectors i.e., a truncated basis,  or any other shape basis which may also be used as descriptors), a functional map can be computed. So theorem 3.1 will still hold. For example, as mentioned in this theorem, if Q_M and Q_N are singular vectors (eigenvectors) of some descriptor and act as a basis, then the first singular vectors will act as soft clusters for the points in that basis (figure 1), as such the iterative functional map refinement will be matching soft clusters in that basis which is what Theorem 3.1 says.  However, we thank the Reviewer for mentioning this to us, it is a typo and we apologize. This should really be the Us mentioned in the introduction of Theorem 3.1 rather than the Qs used in the proof. We will fix this typo in the final version should our paper be accepted as this in no ways changes theorem.
> - W3: In Eqn.(6), the definition of A_M, k(T, T^{k-1}) is again undefined.
> - RW3: We please refer the reviewer to lines 167-172 and 173-178 where  we defined A_M, A_N, A_{M,k} and A_{N,k }.The (T, T^{k-1}) in A_{M,k}(T, T^{k-1}) is trying to recover the new map T^t that aligns the columns given the previous map T^t-1 that aligned the rows. We apologize that since we already mentioned in lines 64-70 we did not see the need to repeat thjs. We will edit to make it explicit here in equation 6 too then.
> - W4: The proof of Theorem 4.1 is pointless. Please use the standard mathematical derivation, rather than piling up a series of unverified statements.
> - RW4: We are sorry to disagree with the Reviewer’s remarks that these statements are unverified. All these statements are mathematically sound and are verifiable to the best of our knowledge. Else, we would never make them for a conference such as Neurips. Statement 1 gives a sound interpretation of the rows of the neighborhood matrices A_{M,k} and A_{N,k} as descriptors of the nodes (neighborhoods as descriptors). Statement 2, states mathematically the role of T^t-1 in A_{M,k}( :, T^t-1 ), that is to align its columns (that is reorder the neighborhood/k-hop descriptor of each node). Statement 3 gives the mathematically verifiable statement that each entry K(i,j) in the product K=A_{M,k}(:,Tt−1)A_{N,k}, will be the number of vertices that are common in the k-hop neighborhoods of vertices i and j, after the alignment of the k-hop neighborhoods as mentioned in statement 2. Thus Statement 4 logically concludes that finding T via equation 6 is equivalent to seeking to match all the vertices i, and j whose k-hop neighborhoods have most vertices in common based on the alignment T^t−1.
>
> # Questions:
> - Q1: I would disagree with the argument in L27. Shape matching methods would rarely aim for solely smoothness without accuracy -- the latter is the most important, while the former can serve as good regularization.
> - RQ1: We refer the Reviewer to our definition of accuracy as the %Correspondences as geodesic error 0. To the best of our knowledge, besides 2D-GEM, Dual Iterative Refinement and GRAMPA, most previous works do not aim for 100% Correspondences at Geodesic error 0 but rather to have this 100% Correspondences at a reasonable geodesic error as seen by even the low % correspondence curves at geodesic error 0 of most of these works. We please refer the Reviewer to 2D-GEM[1] for example who discussed this point extensively.
> - Q2: Should niegh(i) in Eqn.(4) be neigh(i)?
> - RQ2: We apologize for this typing error (typo). We will correct this.
> - Q3: In Fig.4, I can not see the curve regarding HOPE. It would be useful to report geodesic error in the legend.
> - RQ3: This is because HOPE is overlaid by 2D-GEM (zooming in one can see the pink curve of HOPE). We refer the reviewer to lines 297-302 which states “HOPE… achieves 92.54% accuracy at geodesic error 0 on the TOSCA dataset…”. Please see a modified figure in the attached PDF
> - Q4: If I understood correctly, the appendix should be behind checklist.
> - RQ4: We please refer the reviewer to the Neurips paper Checklist requirement at the link https://neurips.cc/public/guides/PaperChecklist.

---

> > ### Comment · Reviewer_E6Vo · 2024-08-07
> > **Let us start with formulation**
> >
> > Still, I do not see a self-contained formulation/explanation regarding Eqn.(1). Could you please kindly refer to a specific formulation in **any** of the referred papers?
> >
> > I got the idea that it is an iterative refinement, it is the undefined notations in Eqn.(1) that makes me confusing and frustrating...

---

> > ### Comment · Reviewer_E6Vo · 2024-08-07
> > **On Theorem 4.1**
> >
> > I have to admit that I have no clue how the proof is going on, as I have trouble to understand the formulation from the beginning.
> >
> > The obvious observation is that the objective of Eqn.(6) is entirely absent from your proof. Where is the Trace going?

---

> > > ### Author Response · Authors · 2024-08-07
> > > **Response to Reviewer E6Vo on "On Theorem 4.1"**
> > >
> > > Again, we thank the Reviewer for seeking more clarity from us. In hope to change the Reviewer’s comment we answer as follows: Equation 6 is the well-known Linear assignment problem where we try to find the permutation matrix T^t which maximizes the assignment of rows to columns (trace maximization). This assignment of rows to columns of a cost matrix where the rows are vetices of shape 1 and the columns are vertices of shape 2 is shape matching. And so our theorem 4.1 shows that trying to do shape matching where our cost matrix is the product of corresponding k-hop neighborhoods (given an initial neighborhood aligning map T^t-1) is the same as finding the map T^t that matches nodes with the most nodes in common in their k-hop neighborhood.

---

> ### Comment · Reviewer_E6Vo · 2024-08-07
> **On the definition of accuracy**
>
> A quick question: how does your definition of accuracy (i.e., %Correspondences as geodesic error 0) make sense at all? For instance, I have a map with 50% points at error 0 + 50% points at error 0.1, and a map with 50% points at error 0 + 50% points at error 1.0, the two are equally good under your definition, is that sensible?
>
> Beyond that, how can you set the percentage of perfect matching as an optimization goal? It is discrete...

---

> > ### Author Response · Authors · 2024-08-07
> > **Response to Reviewer E6Vo on "On the definition of accuracy"**
> >
> > We once again thank the Reviewer for this question. We point out that the Reviewer’s scenario has to do with smoothness. Accuracy as from the definition we adopted from the Neurips paper 2D-GEM has to do with matching at geodesic error 0 strictly, while smoothness has to do with matching as we deviate slightly away from geodesic error 0 (i.e., conservation of neighborhoods). We showed that aiming for accuracy while using k-hop neighborhoods (our HOPE) will not only account for accuracy, but also smoothness in our experiments (hence addresses your scenario).   For example, even when shapes do not have consistent triangulations, we showed that HOPE outperforms baselines at matching away from geodesic error 0. This can be seen in the experiments on the re-meshed shape datasets SCAPE_r, FAUST_r, TOSCA_r , and on partial shapes in the paper’s appendix (and the experiment on the non-isometric SMAL_r in our Rebuttal PDF). In all these our proposed HOPE performs comparatively better than other baselines. An explanation on why using k-hop neighborhoods to enforce accuracy also enforces smoothness has to do with the definition of smoothness itself (which in a simplified way can be defined as a preservation of neighborhoods, a fact known also in the field of graph neural networks).

---

> ### Author Response · Authors · 2024-08-07
> **Response to Reviewer E6Vo on "Let us start with formulation"**
>
> We thank the Reviewer for being willing to ask us for more clarity. In hope of changing the Reviewer’s scores we respond as follows. For equation (1) of our paper we refer the Reviewer to equations 13, 14, 15 and 16 of the CVPR paper “Efficient Deformable Shape Correspondence via Kernel Matching”. Or the ICLR paper. " DEEP GRAPH MATCHING CONSENSUS" in equation 6. Moreover, equation (1) can still be obtained from papers such as the Neurips paper   2D-GEM (“Exact Shape Correspondence via 2D Graph Convolution”), by explicitly writing out their Cp in algorithm 1 using their definition of Cp in equation 10 and Zl in equation 9. This formulation is true for all iterative refinement algorithms that use pairwise descriptors.

---

> ### Comment · Reviewer_E6Vo · 2024-08-07
> **Could you please respond with mathematical formulations?**
>
> I have read all responses regarding formulation, accuracy, and Theorem 4.1. And I am still confused, could you please provide answers with self-contained, well-defined derivation to my questions, instead of a pile of sentences?
>
> For instance, in kernel matching paper, everything is expressed as clear matrix operation, while in your case, you say something like this:
>
> 'W_M(T^t, T^{t−1}) \in R^{n×n} is the pair-wise descriptors of vertices for shape M with its rows and columns aligned using the map T^t and the previous iterations map T^{t−1}'
>
> I have not doubted the correctness of the formulation. The problem is that the above definition is wordy and confusing. How is W_M computed/defined explicitly?
>
> Similarly, I can imagine there is a link between the trace and assignment problem. However, your response again is descriptive. A much better answer would be to explicitly draw connection between such with **equations**.
>
> On accuracy, the impression I got is that the accuracy you focused on is ultimately not different from the general accuracy (i.e., mean geodesic error), as shown in Fig. 4 and 5. Also an obvious fact is that defining accuracy as percentage of perfect matching is not optimizable in practice -- as the ground-truth annotation is not always available, in fact, most of the time.
>
> Though I would be happy to discuss and open to be corrected, I would expect efficient, precise communications rather than reading *hand-wavy arguments* and deriving *claimed equivalent forms* myself.

---

> > ### Author Response · Authors · 2024-08-08
> > **Response to Reviewer E6Vo on "Could you please respond with mathematical formulations?"**
> >
> > We apologize to the Reviewer if our answers to the Reviewer’s concerns were not clear.  With hope to change the Reviewer’s scores, please find attached responses in bullets below with as much formulations as possible. We hope they will help with more clarification:
> > - W_N \in T^{n * n} is a matrix of pairwise descriptor for vertices n of shape N. It can be geodesic distances or any other pairwise descriptors (In our case we use k-hop neighborhoods as seen in lines 166-202 of our paper)
> > - T^t and T^{t-1} are the current map and the previous map we want to recover and are \inR^{1*n}. For example, if we have two shapes M and N with 4 vertices each and the map T^{t-1} = [ 2, 1,0, 3], then this means vertices 0,1,2,3 in shape N correspond to vertices 2,1,0,3 (respectively) in shape M according to this map.
> > - The Trace in the Linear Assignment Problem (LAP): The LAP involves finding a permutation (P) of the columns of a cost matrix ( C ) such that the sum of the corresponding diagonal elements (the trace of the permuted matrix) is minimized or maximized. It is formulated as argmax_P Trace(PC). In our case our cost matrix is C = W_N(:, T^{t-1})*W_M = W_N*P’W_M (where P’ = mat(T^{t-1}) i.e., the matrix form of T^{t-1}). And our LAP is argmax_P Trace(PC) where P = mat(T^{t}) i.e., the matrix form of T^{t}).
> > - For accuracy, indeed directly enforcing this constraint in the optimization problem is impossible as the Reviewer rightly said because the ground truth map is not available. As such the accuracy constraint is usually enforced by trying to use accuracy enforcing pairwise or point-wise descriptors as we discussed in Section 3 of the paper.

---

> ### Comment · Reviewer_E6Vo · 2024-08-12
> **Some final remarks**
>
> For the sake of fairness, I have re-examined this submission with the clarification made by the authors. Below are my new questions:
>
> 1.	On Theorem 3.1:
>
> Let alone the confusing purpose of this theorem in the **related work** section, I am confused by the theorem description and the proof. Following the authors’ rebuttal, W in Eqn.(1) denotes pair-wise descriptor such as geodesics, Q in Eqn.(2) denotes vertex-wise descriptor such as SHOT, HKS.
> The claim of theorem 3.1 is “Using **W** for the map refinement via Functional maps helps group nearby clusters together assuming the functional map is perfectly accurate. “. On the other hand, in the proof, **Q** is used throughout. To the best of my knowledge, these two should be assumed to be independent (except for cases where Q is derived from W), right?
>
> 2.	On Theorem 4.1:
>
> In general, a serious theoretical analysis for map refinement method should include at least two perspectives: 1) the proof of convergence; 2) the proof of certain property. For example, Zoomout proves that the (ideal) optimal solution is an isometric map between two shapes. Huang and Guibas [a] prove that their algorithm can recover exact maps under certain noise model (with respect to initial maps).
>
> Regarding this submission, I am not sure what is the theoretical insight about Theorem 4.1: The claim is about agreement with T^{t-1}, which can be traced back to initialization. As I mentioned above, the expected theoretical analysis should be something like the algorithm can converge to good solution even if initialization is not perfect. To this end, arguing some property aligning with previous iterations seem point-less to me.
>
> [a] Consistent Shape Maps via Semidefinite Programming, Q. Huang and L. Guibas, SGP, 2013.
>
> 3.	On connectivity
>
> The submission claims superior performance in TOPKIDS. In Fig.2, the error plot is nearly perfect, however the qualitative result seems to be worse than Zoomout. Also, if the method depends on adjacency, then how can it be robust with respect to topological noises, i.e., there are a lot of short cuts in the regarding meshes?
>
> 4.	On experimental results
>
> The performance gap between SCAPE and SCAPE_r (Fig.4(b) vs. Fig.5(b)) seems to suggest the proposed method to some extent depends on the unified triangulation prior, which is not promising as the shape matching community has long passed the point caring about matching shapes with uniform triangulation.
>
> To be honest, I can go on with the question list, but my time could have been contributed to something more meaningful. I have tried my best to understand and evaluate this submission, but can hardly find any evidence for changing my initial rating.

---

> ### Author Response · Authors · 2024-08-12
> **Responses to "Some final remarks" by Reviewer E6Vo**
>
> Dear Reviewer, we are grateful for the time spent to Review our work. We address the concerns and misunderstandings raised in the final remarks below:
>
> **On Theorem 3.1:**
>
> In our response to the Reviewer’s first Reviews we addressed this by pointing out to the Reviewer that we had made a typo here in the proof of the theorem by using Q instead of U as in the definition of the theorem. But this does not change the proof nor the conclusion of the theorem. This theorem proves that all iterative map refinement methods that use truncated basis (such as first k-eigen vectors or singular vectors of some shape descriptors whether pairwise or pointwise) will essentially try to refine the map by aligning clusters of points where with clusters being defined by this truncated basis (since it is well known that the truncated SVD or eigen decomposition are soft clusters). Thus, all these methods will fail at accuracy since it will be matching clusters rather than points. While to some extent in many cases maintaining smoothness depending on which descriptor the truncated basis was obtained from.
>
> **On Theorem 4.1:**
>
> Regarding the proof that the k-hop algorithm will recover the ground truth map under certain conditions, we refer the Reviewer again to Section 4.2, particularly lines 167-175 where we cited [26] and [48] which both give riguouruous and lengthy proofs for using 1 hop or k hop adjacencies for refinement. We then gave Theorem 4.1 that shows using k= 1, 2,3, …, hops for is same as using different hop neighbors as witnesses for the map refinement. As such, we propose to use different hop neighborhoods iteratively which will serve to enforce both local and global consistency in the map (See Section 4.2-4.4).
>
> **On connectivity**
>
> We do not only claim superior performance to ZoomOut but prove this experimentally on most if not all datasets by a hudge margin in some cases. The curves show this as pointed out by the Reviewer. For the Qualitative analysis. We refer the Reviewer to figure 4 in our attached PDF as response to all Reviewers. It clearly shows that ZoomOut fails to be accurate or even be continuous.
>
> **On experimental results**
>
> We refer the Reviewer to the variety of datasets used again and point out that on all these datasets including the ones where the triangulations are not consistent, HOPE (our work) outperforms all baselines in some cases by a significant margin.
> We admitted that HOPE is limited because it relies on the triangulations (in our limitation Section 6). Nonetheless, we showed that even on non-correlated meshes HOPE still outperforms other baselines.
>
> **Additional Remarks**
>
> Given all these, we really believe our work brings a significant contribution to the shape matching community and welcome any further questions from the Reviewer. Moreover, we provided the Source code for HOPE as supplementary material which the Reviewer may use to test other custom meshes (and feel free to give us feedback).

---

### Official Review · Reviewer_dry5 · 2024-07-11

**Soundness:** 3
**Presentation:** 3
**Contribution:** 3
**Rating:** 7
**Confidence:** 4

**Summary:**

The paper focuses on the shape-matching tasks and proposes a shape-matching refinement technique based on the K-hop neighborhood descriptor and local map distortions. The experiments show improved results compared to existing approaches in isometric and non-isometric shape registration.

**Strengths:**

- The paper presents the background, theory, and methodology very well.
- The methodology is simple and sound and can positively impact shape-matching refinement.
- The performance, runtime, and generalizations are promising.

**Weaknesses:**

- While there is existing work on deep and modern feature extraction, the experiments in this paper are limited to the SHOT descriptor. Incorporating other classical or learned signatures could demonstrate broader generalizable performance. Additionally, evaluating the impact of the quality of initial matches could provide further insights into the robustness of the proposed method.

- The paper is motivated by the observation that existing approaches often prioritize smoothness over accuracy. Although this paper emphasizes improving accuracy, it raises the question of whether smoothness might be compromised as a result. Detailed analysis and results are needed to assess the balance between these two aspects.

**Questions:**

The paper suggests using geodesic error as a loss function to improve smoothness and accuracy. Given that other methods also utilize geodesic error, what aspects of the HOPE approach allow it to leverage geodesic error more effectively and potentially yield superior results? How does this approach manage the balance between local and global geometric consistency?

**Limitations:**

The method might face challenges in handling topological variations between shapes.

---

> ### Author Rebuttal · Authors · 2024-07-31
>
> We thank the Reviewer for the insightful and constructive comments. We address some of the Reviewer’s comments and suggestions below:
>
> # Weaknesses:
> - W1: While there is existing work on deep and modern feature extraction, the experiments in this paper are limited to the SHOT descriptor. Incorporating other classical or learned signatures could demonstrate broader generalizable performance. Additionally, evaluating the impact of the quality of initial matches could provide further insights into the robustness of the proposed method.
> - RW1: We refer the reviewer to Appendix C where we equally use the HKS and WKS as initializations. We added more qualitative results in the attached PDF to the global response and will add them to the appendix as well.
> - W2: The paper is motivated by the observation that existing approaches often prioritize smoothness over accuracy. Although this paper emphasizes improving accuracy, it raises the question of whether smoothness might be compromised as a result. Detailed analysis and results are needed to assess the balance between these two aspects.
> - RW2: Mathematically, an accurate map will equally be smooth (in case the map recovered is same as the ground truth for example). But as the Reviewer points out, since most models may not perfectly recover this ground truth map, there will be a tradeoff between aiming for smoothness and accuracy as discussed in Section 3 which shows the difficulty to maintain smoothness while aiming for accuracy by methods such as such SHOT, since it does not strictly enforce smoothness. A more precise and general response will be that the level of smoothness of the final map obtained by an algorithm (aiming for accuracy) will depend on how it enforces this smoothness constraint as while aiming for accuracy. For example, in the case of HOPE, one can observe that enforcing local and global neighborhood consistency in order to aim for accuracy also enforces the smoothness constraint since points will be matched if their local and global neighborhoods are similar. We will add this discussion in the Appendix.
>
> # Questions:
> - Q1: The paper suggests using geodesic error as a loss function to improve smoothness and accuracy. Given that other methods also utilize geodesic error, what aspects of the HOPE approach allow it to leverage geodesic error more effectively and potentially yield superior results? How does this approach manage the balance between local and global geometric consistency?
> - RQ1: We used the geodesic error as a measure of accuracy (via measuring the %Correspondences at geodesic error 0) and smoothness (by seeing how rapidly the %Correspondence increase as we move away from geodesic error 0).  In order to achieve this accuracy we employed the concept of neighborhoods to refine an initialized map. We used both local (lower hops) neighborhoods and global (higher hops) neighborhoods to match points. When the local and global neighborhoods of points were similar (based on some initial map) we matched them together. The reviewer indeed raise an interesting future research question on how to balance the importance of local consistency with that of global consistency. We briefly discussed this in Section 3 where we highlighted the reasons on why GRAMPA failed in matching isometric shapes where the local neighborhoods of the meshes are similar. It has been studied in graph isomorphism (with models such as the Wesfeiler Lehman), and even in the field of Graph Neural Networks (in papers such as “How powerful are graph neural Networks” or “Topological graph Neural Networks”) that local neighborhoods being similar does not necessarily imply a match. For example, if in the neighborhood of node A, two of its neighbors are white and two are black, and in the neighborhood if node B the same class distribution is observed, even if the connectivity in these neighborhoods are the same, it does not necessarily mean that these two nodes are the same as their 2 hop neighbors may be significantly different. We will add discussion on this in the main paper or the appendix as well.
>
> # Limitations:
> - L1: The method might face challenges in handling topological variations between shapes.
> - RL1: Indeed the main limitation of our work is that we rely on the mesh connectivity as mentioned in Section~6

---

> > ### Comment · Reviewer_dry5 · 2024-08-12
> > **Responce to rebuttal**
> >
> > Dear authors,
> >
> > Thank you for submitting your rebuttal and addressing the concerns raised. I have reviewed the rebuttal and found that my questions have been addressed.
> >
> > Best regards,
> > dry5

---

### Author Rebuttal · Authors · 2024-08-02

# Comment To All Reviewers
- First, we thank the Reviewer for their comments and please direct each Reviewer to specific responses to each as well as to the general responses here and the attached PDF.
- We added experiments on intra-class matching on the SMAL_r dataset as suggested by Reviewer NAeg. We hope it convinces the Reviewer about the quality of our work, as this experiment further demonstrates the choice of HOPE over existing baselines. We used the same parameters as on other datasets in the main paper since one Key benefit of HOPE is that we do not need any parameter tweaking. For this experiment, we match 298 intr-class shape pairs and report he average geodesic curves of HOPE, 2D-GEM, DIR, and ZoomOut.
- We added some Qualitative visuals on the maps from different baselines as suggested by Reviewer dry5. However, due to space we could not add on other datasets besides the three chosen. We will add more to the main paper in the appendix in case of acceptance
- We will equally add all related works suggested by all Reviewers to the main paper in case of acceptance.
- We apologize for any miss understanding with Reviewer  E6Vo and plead for a look at our rebuttal as well as ask us any further questions as we really hope to address all concerns and convince the Reviewer about the quality of our work.

---

### Decision · Program_Chairs · 2024-09-25

**Decision:**

Accept (poster)

**Comment:**

The paper obtained mixed reviews, with two reviewers voting for reject and two voting for accept. On the positive side, good presentation, theoretical contribution (simple and sound methodology) and good performance are noted. On the downside, limited novelty, limited evaluation, and a negative trade-off with smoothness of the resulting correspondence. The extensive rebuttal is acknowledged. In particular, we weighted down one very negative review in light of its divergent complaints regarding the presentation and lack of understandability that could not be corroborated. We urge the authors to still improve the paper regarding the raised concerns, though. Overall, we follow the positive reviews and the paper should be accepted in light of its theoretical contribution and practical advantages.